# Effect of Laser Auriculotherapy on Quality of Life, Fatigue, and Anxiety in Women with Advanced Breast Cancer: Randomized Clinical Trial

**DOI:** 10.3390/healthcare13020166

**Published:** 2025-01-16

**Authors:** Larissa Marcondes, Poliana Martins Ferreira, Namie Okino Sawada, Tania Couto Machado Chianca, Jorge Vinícius Cestari Felix, Paulo Ricardo Bittencourt Guimarães, Luciana Puchalski Kalinke

**Affiliations:** 1Department of Nursing, Federal University of Paraná (UFPR), Curitiba 81210-170, Brazil; jvcfelix@hotmail.com (J.V.C.F.); kalinkeluciana@gmail.com (L.P.K.); 2Department of Nursing, Federal University of Alfenas (UNIFAL), Alfenas 37130-001, Brazil; poliana.martins@sou.unifal-mg.edu.br (P.M.F.); namie.sawada@unifal-mg.edu.br (N.O.S.); 3Department of Nursing, Federal University of Minas Gerais, Belo Horizonte 30130-100, Brazil; taniachianca@gmail.com; 4Department of Statistics, Federal University of Paraná (UFPR), Curitiba 81531-990, Brazil; guimaraes.prb@gmail.com

**Keywords:** auriculotherapy, breast cancer, complementary therapies, oncology nursing, quality of life

## Abstract

**Objective**: To evaluate the effect of laser auriculotherapy on health-related quality of life, cancer-related fatigue, and anxiety in women with advanced breast cancer undergoing palliative chemotherapy. **Method**: A randomized, parallel, single-blind, single-center clinical trial was conducted in a Brazilian oncology referral hospital. A total of 123 women were randomly divided into groups: 41 in the experimental group (10 weekly laser auriculotherapy sessions), 40 in the sham group (10 weekly sham sessions), and 42 in the control group. Outcomes were assessed at baseline, 5th week, and 11th week, using the Quality of Life Questionnaire Core-30, Functional Assessment of Cancer Therapy: Fatigue and State-Trait Anxiety Inventory. Statistical analyses included Kruskal–Wallis, Mann–Whitney, and Friedman tests. **Results**: The experimental group exhibited the highest mean overall quality of life, with a significant increase (*p* < 0.000001) between the second and third evaluations and a significant improvement in fatigue between the second and third evaluations (*p* = 0.00001). Conversely, women in the sham and control groups experienced a decline and worsening in both their quality of life and fatigue. Women in all three groups showed reduction an anxiety. Changes in anxiety in these women were not statistically significant. **Conclusions**: reduction improvement in health-related quality of life and a reduction in cancer-related fatigue in the experimental group. These results show a positive effect of auriculotherapy on cancer-related fatigue, demonstrating potential for application in clinical practice in women with advanced breast cancer and in palliative chemotherapy. Brazilian Registry of Clinical Trials: RBR-6hxffx4.

## 1. Introduction

Breast cancer was responsible for the highest cancer incidence and mortality in women in 2020, with approximately 2.3 million new cases and 684,996 deaths [1]. Estimates published for 2023 in Brazil represent an incidence rate of 41.89 new cases per 100,000 women, and a death rate of 11.84 per 100,000 women in 2020 [2]. Advanced breast cancer (ABC) is responsible for 90% of deaths, with 20 to 30% of cases being early-stage breast cancer and the rest due to cancer progression [3].

Palliative therapy is recommended for women diagnosed with ABC. It involves systemic treatment to improve symptoms, control the disease, manage side effects, and contribute to the improvement of quality of life [4]. Chemotherapy (CT) and hormone therapy (HT) are the primary palliative therapies. However, they induce physical symptoms such as fatigue, psychological discomfort, maladaptive coping, and anxiety, contributing to a worsening of survival [5].

Fatigue and anxiety negatively affect activities of daily living and have a substantial impact on health-related quality of life (HRQoL), which is understood as the individual’s perception of their health and its impact on one’s life [5,6]. Cancer-related fatigue (CRF) is a complex, multidimensional symptom with a significant impact on HRQoL. It is “a distressing, persistent, subjective sense of physical, emotional, and/or cognitive tiredness or exhaustion related to cancer or cancer treatment, that is not proportional to recent activity; and interferes with usual functioning” [7]. It affects 80–99% of patients on chemotherapy [8]. Anxiety is an unpleasant emotion resulting from intrusive feelings about an uncertain future that results in psychological and physical manifestations, with different consequences and negative impacts on HRQoL [9].

Fatigue associated with anxiety in women with ABC during chemotherapy may lead to maladaptive coping and psychological distress. In these conditions, 60.2% of women may experience anxiety, 52.3% depression, and 36.9% stress due to the uncertainty and risk of mortality that accompany the disease. These results demonstrate the necessity for the health team to be well-equipped to provide diverse therapeutic approaches addressing both physical and emotional challenges faced by these women [5].

The use of non-pharmacological therapies (NPT) is one approach that may be indicated for the management of CRF and anxiety. They are used to prevent or complement conventional treatment, are based on theories focused on the bio-psychosocial aspects of the individual facing the health-disease process, and can contribute to the improvement of HRQoL [10].

Auriculotherapy is a type of NPT. It is an ancient acupuncture technique involving the stimulation of specific points called acupoints, located in the pinna, using needles, magnets, spheres, mustard seeds, crystals, or lasers [11,12,13]. It is an effective clinical practice in the control of the multiple effects of the disease and in oncological treatment. It can be used as an isolated approach or integrated with conventional treatments [14,15]. In recent decades, research in the area of auriculotherapy [16,17,18] has adopted the use of low-power lasers, instead of needles, as a method of stimulating the auricular acupoints. This approach aims to minimize the invasiveness and pain associated with the therapeutic process, making it more acceptable to patients. However, studies using laser auriculotherapy (LA) as a complementary treatment for fatigue and anxiety in cancer patients are lacking.

As NPT can minimize the effects of palliative therapy (fatigue and anxiety) and help improve HRQoL, this study aims to evaluate the effect of LA on cancer-related fatigue, anxiety, and health-related quality of life in women with ABC undergoing palliative chemotherapy.

## 2. Materials and Methods

The study design was guided by the recommendations of the Consolidated Standards of Reporting Trials (CONSORT), and followed the Standards for Reporting Interventions in Clinical Trials of Acupuncture (STRICTA) tool.

The study was an open randomized, parallel, 1:1:1, single-blind, single-center, clinical trial, including women aged 18 and above diagnosed with ABC. The participants started the first or second line of palliative chemotherapy, screened by the Authorization for High Complexity Procedure (APAC) code of the Unified Health System (SUS), a public health system in Brazil that guarantees universal, full, and free-of-charge access to health services. Hospitalized patients and/or those using other NPTs were excluded.

The study considered several discontinuity criteria, including patients who refused to continue participating in the intervention and/or complete the questionnaires, gave up conventional treatment or interrupted treatment, failed to attend two consecutive sessions for personal reasons, had their chemotherapy treatment suspended, or died.

The women were recruited from the outpatient chemotherapy service of an oncology hospital, a national referral hospital in southern Brazil. First, the electronic medical records and APAC records of all patients scheduled to start treatment were analyzed. Eligible patients were invited to participate and included in the study protocol after signing an informed consent form before their first palliative chemotherapy session.

Participants in the three groups underwent a standard scheduled treatment: palliative chemotherapy followed up by the multidisciplinary team according to the institution’s periodicity protocol. The experimental group (EG) received the intervention (laser auriculotherapy technique), the sham group (SG) received the simulation of the technique (auriculotherapy without laser emission), and the control group (CG) received only the standard treatment. All groups were followed up for 11 weeks.

The EG received the intervention with LA, which was applied in 10 sessions, once a week, for 10 weeks. The participants were attended in a private room, comfortably positioned on a stretcher, and black glasses were placed to protect them from the laser and blind the group. Their ears (right and left) were treated with antiseptic using cotton wool and 70% alcohol, with subsequent application of the intervention. The laser used was the Therapy ilib^®^ model [19], DMC brand, a wireless device with a 100 mW red laser, which was placed in direct contact with the skin of the pinna, at a power of two joules (2J) per acupoint. This is the most recurrent dosage among the articles presented by the systematic review on the clinical efficacy of laser acupuncture, with a positive result [16,17,18,20]. Treatment continued for 20 s at each of the following auricular points (Figure 1): Central Nervous System or Shen Men, Kidney, and Autonomic Nervous System, which are characterized as “opening points”, due to their role in opening or unblocking energy channels in auriculotherapy and promoting the proper flow of vital energy in the body.

For anxiety, the following points were used: brain stem, spleen, anterior anxiety, and diaphragm, which together reduce anxiety, act as an analgesic and sedative, and calm the mind and spirit. In addition to muscle relaxation, sleep, lung, spleen, and adrenal points, they were used to help improve body functioning and reduce cancer fatigue. Point stimulation followed the order described above. The dominant side of the participant was considered for the choice of ear, except the spleen, in which the stimulus was always performed in the left ear due to the anatomical location of the organ in the human body [21]. At the beginning of each session, starting from the second session, the patients were asked if the application of the laser in the previous week led to any discomfort.

Simulated auriculotherapy (Sham) was administered to the SG. Participants in this group were blinded to the EG, i.e., they were unaware whether they received sham auriculotherapy or laser therapy. Participants underwent ten sessions of simulated auriculotherapy (one per week), without any laser output and/or stimulation, the laser being off during the simulation of the technique. The acupoints, duration of application, and total number of applications were equal to the EG. The auricular mapping followed the location references of the Atlas of Auriculotherapy [21].

The technique for the EG and SG lasted approximately 10 min. Both groups wore black protective glasses, impeding the visualization of the laser light emission. All sessions were conducted with the participants face-to-face, once a week, by the researcher with a degree in nursing, improvement, qualification, and specific training for the auriculotherapy technique and 3 years of experience in the application of the technique, to ensure the uniformity of the protocol. The equipment was sanitized, according to the service’s cleaning and disinfection routine, before and after use for each participant.

CG participants did not undergo any intervention. The participants were only requested to complete the evaluative questionnaires.

The participants comprised 123 women with ABC on first- or second-line palliative chemotherapy. The sample size calculation (n = 123) was performed by simple dimensioning, non-probabilistic sampling, based on the mean (150 new cases) of patients who underwent first- or second-line palliative chemotherapy for advanced breast cancer from 2018 to 2020, with a margin of error of 3.8% and a 95% confidence level.

The randomization process was carried out by drawing lots, by blocks, with envelopes containing numbers from 1 to 30, each numbering sequence corresponded to the allocated group (Table 1), respecting the allocation at the ratio of 1:1:1. The envelopes were sealed after the numbers were placed. When they signed the informed consent form, patients were instructed to draw a card from inside the envelope, open it, and check which group they belonged to.

The primary outcome of this study was the reduction in side effects from palliative chemotherapy and improvement in the HRQoL score of ABC patients. These were measured using the Quality of Life Questionnaire Core-30 (QLQ-C30) in the group of patients who underwent LA. As a secondary outcome, a reduction in anxiety levels was assessed, measured through the State-Trait Anxiety Inventory, and an improvement in fatigue indices, measured by the Functional Assessment of Cancer Therapy: Fatigue questionnaire.

All groups (EG, SG, and CG) answered a questionnaire with sociodemographic and clinical variables, the QLQ-C30, Functional Assessment of Cancer Therapy: Fatigue, and the State-Trait Anxiety Inventory at three different stages.

The researchers of the study developed a sociodemographic and clinical questionnaire (SCQ), which has already been used in previous studies and comprises 29 items, divided into two parts: sociodemographic characterization and clinical characterization. The sociodemographic questions refer to the following variables: age, gender, city of origin, family member, area of residence, race/ethnicity, marital status, number of children, education, profession/occupation, family income, National Institute of Social Security (NISS) benefit, government allowance, religious belief, and religious practice. The clinical characterization items were diagnosis, date of diagnosis, clinical staging, presence of metastasis, previous treatments, family history of cancer, clinical comorbidities, smoking, alcohol consumption, use of continuous medication, chemotherapy protocol, and Karnofsky performance status.

The QLQ-C30, developed by the European Organization for Research and Treatment of Cancer (EORTC), was used to assess HRQoL. This instrument is composed of 30 questions, divided into four domains: functional scale (15 questions); symptom scale (7 questions); simple items (6 questions); and overall QoL (2 questions). Questions 1 to 28 are measured according to a Likert scale from 1 to 4 (1—not at all, 2—a little, 3—quite a bit, 4—very much), and questions 29 and 30 are expressed on a seven-point semantic differential scale. The sum of the items can generate a score ranging from 0 to 100. For the functional scales and global health status/QoL, the higher the score, the greater the functionality and quality of life. For the symptom scale, in contrast, the higher the score, the greater the indication of a high level of symptoms/problems [22].

Fatigue was assessed using the Functional Assessment of Cancer Therapy: Fatigue (FACIT-F) (Version 4), a 40-item instrument translated and validated for Brazilian Portuguese [23]. This questionnaire has a section related to global quality of life, the functional assessment of cancer therapy: general (FACT-G), with twenty-seven items, subdivided into four other main domains (physical [seven questions], social/family [seven], emotional [six], and functional [seven]), aiming to analyze all dimensions of the cancer patient, and 13 specific items in additional concerns that refer to fatigue (fatigue subscale), all of them measured by a Likert scale from 0 to 4. The score of the instrument is obtained for each domain through the means of the answers, and the total score is obtained through the sum of the scores of the domains. The FACT-G score can range from 0 to 108. The higher the final score, the better the quality of life. Regarding the subscales of additional concerns, the FACIT-Fatigue (subscale) scores range from 0 to 52, and the higher the final score, the lower the fatigue. The Trial Outcome Index (TOI)—composed of the sum of physical well-being, functional well-being, and additional concerns—ranges from 0 to 108 [23,24]. The HRQoL scores obtained with the FACIT-F allow the assessment of physical, social/family, emotional, and functional well-being; FS; TOI; FACT-G; and the evaluation of the instrument as a whole (FACIT-F = FACT-G + FS).

To assess anxiety, we used the State-Trait Anxiety Inventory (STAI-S; STAI-T), an instrument [25] that was translated and adapted to Brazilian Portuguese [26]. It consists of two self-report scales that assess anxiety as a state (STAI-S) or trait (STAI-T). Each situation (state and trait) has 20 items with scores from 1 to 4 in each. The score ranges from 20 to 80. The population mean is 40, >42 tend to have anxiety, and <38 tend to have depression.

Three stages were defined for data collection in this study: T1, T2, and T3. All instruments (QSDC, QLQ-C30, FACIT-F, STAI-S, and STAI-T) were applied to the three groups at baseline (first week), taken as T1. At follow-up, T2 and T3 (fifth and eleventh week, respectively), the QLQ-C30, FACIT-F, and STAI-S questionnaires were reapplied. In the EG and SG, the application occurred before the intervention, and in the CG after the conventional treatment. The intervention and data collection were conducted in a private room, with participants spending 8 to 13 min to complete the instruments.

The collected data were independently duplicated, validated, and stored electronically in Microsoft Excel Office 365^®^ spreadsheets. The results of the sociodemographic and clinical characterization underwent descriptive analyses, including distribution, absolute frequency, relative, mean, standard deviation, and variation. The analysis of HRQoL, fatigue, and anxiety data was performed using the software Statistica^®^, Version 7. The assessment of the bivariate relationship between the categories of clinical and sociodemographic variables and HRQoL, fatigue, and anxiety scores was verified by applying the Chi-square test, Kruskal–Wallis test, and Mann–Whitney test. The comparison between the evaluation moments was performed using the Friedman test. To complement the significant results obtained by the Kruskal–Wallis and Friedman tests, the least significant difference (MSD) test was applied. The lack of normality of the data was proven using the Shapiro–Wilk test. Spearman’s correlation coefficient was used to verify the correlation between health-related quality-of-life measures and cancer-related fatigue.

The study complied with Resolution 466/12 of the National Health Council of the Brazilian Ministry of Health, which regulates ethical guidelines for research involving human beings. It was approved by the Research Ethics Committee of the study setting with Opinion No. 4.704.263 of 11 May 2021. The clinical trial was approved (RBR-6hxffx4) by the Brazilian Registry of Clinical Trials.

## 3. Results

### 3.1. Recruitment, Withdrawal, and Retention of Participants

Of the 126 adult women who initiated first- or second-line palliative chemotherapy for ABC in the study period, 3 refused to participate, primarily due to an unwillingness to attend the hospital for weekly therapy. Therefore, 123 women were randomized to three groups (EG: 41/SG: 40/CG: 42). During the RCT, three women succumbed to the disease after the 5th week of follow-up. Consequently, 120 women completed all phases of the survey, resulting in a retention rate of 98% (120/123). Recruitment and follow-up spanned from September 2021 to October 2022 (Figure 2).

### 3.2. General Characteristics and Homogeneity of Participants

The sociodemographic and clinical characteristics of the EG, SG, and CG were similar regarding marital status, income, number of children, religious beliefs, and number of metastases (Table 2). However, the groups were noted to be non-homogeneous, with a significant difference in educational levels—lower in the SG. Regarding employment status, the CG had more unemployed women, whereas the SG had more active participants than the others. Additionally, differences were observed based on the presence of comorbidities, with the women in the EG and SG having more comorbidities than those in the CG. There were no statistically significant variations in the performance of functional disabilities or deficiencies (Karnofsky). The means approached 70% and 69% across both groups, demonstrating equivalent performance among the three groups.

### 3.3. Effects of Laser Auriculotherapy

When HRQoL was assessed using the QLQ-C30, it was observed that the baseline overall quality of life (OQoL) of all patients in all groups showed a mean value of 28.76/100 (EG = 31.71/100, SG = 28.85/100, and CG = 25.79/100, respectively), with the Kruskal–Wallis test comparing the groups yielding a *p*-value of 0.053. This indicates that patients perceived their HRQoL as low (poor) before starting palliative chemotherapy and that the groups were homogeneous regarding this variable. The scores of the QLQ-C30 instrument revealed that the highest means of overall quality of life (OQoL) were observed in the EG, however, it is noteworthy that the test between groups demonstrated that the three groups are homogeneous. The EG showed a significant increase (*p* < 0.001) in the score between T2 and T3, demonstrating an improvement in quality of life. The SG and CG showed a significant reduction in OQoL throughout the evaluations, between T1 and T2, and T2 and T3, respectively (SG and CG *p* < 0.001). A significant difference was observed between the EG and SG and between the EG and CG at T2 and T3 (*p* < 0.001), indicating that better HRQoL scores were found in the group that received LA (Table 3).

Concerning physical functioning, the three groups showed oscillations in the scores. Worse scores were found in the groups that did not receive the intervention. The EG displayed improvement between the evaluations while the CG worsened; however, this difference was not significant. In the SG, a statistically significant decline was observed between T2 and T3 (*p* = 0.005). In the comparison between the groups, a significant difference was observed in the physical functioning scores between the EG and SG at T3 (Table 2).

The emotional functioning domain had worse scores in the CG and SG, and better scores in the EG. A statistically significant difference was observed between the evaluations in the EG (*p* = 0.007), with an increase in scores between T2 and T3. At T3, the scores in the EG and SG were significantly different from those between the EG and CG (*p* = 0.004). When cognitive functioning was observed, the EG was stable, and the CG and SG worsened throughout the evaluations, with significant differences between T2 and T3 (*p* = 0.008 and *p* < 0.001, respectively). The social functioning domain showed a significant worsening between T2 and T3 for the EG, SG, and CG (*p* = 0.006/*p* < 0.001/*p* = 0.003, respectively) (Table 2).

According to the symptom scales, nausea and vomiting were significantly increased in the three groups, indicating a worsening between T2 and T3 (EG *p* < 0.001/SG *p* < 0.001/CG *p* = 0.002) and loss of appetite (EG *p* < 0.001/SG *p* < 0.001/CG *p* < 0.001). Meanwhile, pain was significantly increased between T2 and T3 in the SG *p* < 0.001 and CG *p* = 0.012, as well as dyspnea in the SG between T2 and T3, *p* = 0.000. The incidence of diarrhea was decreased between T2 and T3 in the EG (*p* = 0.001) but increased in the CG and SG, although it was not statistically significant (Table 4).

When fatigue symptoms were analyzed, there was a decrease between stages T2 and T3 in the EG (*p* = 0.020) and an increase in the SG and CG (*p* = 0.020/*p* = 0.001, respectively) (Table 3). The results of the Kruskal–Wallis analysis showed that the scores were significantly different between the SG and EG and between the CG and EG (*p* < 0.001) at T3, indicating that fatigue was worse in the groups that did not receive LA.

The results of FACIT-F showed that physical well-being was increased in the EG between T1 and T2 and then stabilized at T3. Meanwhile, in the SG and CG, well-being was increased between T1 and T2 and then decreased between T2 and T3; the difference was significant in the SG (*p* = 0.00005) (Table 5). Despite the increase in scores between T1–T3 in the EG, the results were not significant, and the same was observed for the TOI. When the Kruskal–Wallis test was applied to assess differences between groups, the mean score in T3 for the EG was significantly higher than the other groups for both physical well-being (*p* = 0.004) and the TOI (*p* < 0.001). Thus, it is demonstrated that women in the EG showed an improvement in physical well-being scores after the LA intervention, highlighting that LA is a valuable ally in addressing physical symptoms.

For social/family well-being, SG and CG showed a slight but not significant increase in T2 scores, and the EG showed less variation in the scores, remaining stable (Table 5). When the Kruskal–Wallis test was applied to assess differences between groups, the mean scores in T2 and T3 for the EG were significantly higher than those of the other groups (SG *p* = 0.005; CG *p* = 0.004, respectively).

Regarding emotional well-being, the EG showed an increase in scores at T2 and T3, and the SG and CG showed a decrease at T2 and maintained a decline at T3. No statistical significance was observed for all groups. For this domain, in the comparison between groups conducted using the Kruskal–Wallis test, the third evaluation showed that the EG achieved significantly higher mean scores than the SG and CG (*p* = 0.002). This demonstrates that women in the EG experienced an improvement in emotional well-being over the 11 weeks of LA.

In functional well-being, all groups showed an improvement, with an increase in T2 scores followed by a decline in T3. The same occurred in the TOI in the SG and CG. For the fatigue subscale, the EG showed a statistically significant increase in scores (*p* = 0.017), resulting in less fatigue between T1 and T2, and a decline in the SG and CG scores between T2 and T3, with higher fatigue for these groups. A decrease was observed in FACT-G and FACIT-F scores between T2 and T3 in the SG and CG, with statistical significance for SG (*p* = 0.0005; *p* = 0.003). The EG had better scores, with a significant increase in T2 in both the FACT-G (*p* = 0.002) and the FACIT-F (*p* = 0.011), and remained stable at T3 (Table 5). In the comparison between groups, the EG presented statistically significantly higher FACIT-F scores at T3 (*p* < 0.001). HRQoL scores were better in the EG after the application of LA.

In the FACIT-Fatigue evaluations, there was a significant difference between the groups at T3 (T1 *p* = 0.800; T2 *p* = 0.900; T3 *p* < 0.001), with better fatigue scores in the EG compared to those in the SG and CG. Similarly, a significant increase in fatigue scores was observed in the EG, indicating a reduction in CRF in the second evaluation, between T1 and T2 (*p* = 0.017). In the other groups, no significant change was observed throughout the evaluations (Figure 3). Thus, women in the EG had an improvement in fatigue scores after the first five applications of LA.

The FACIT-F score was significantly positively correlated to the FACT-G and FACIT-F quality of life scores at all stages and in all groups (Table 6). This result indicates that the fatigue of women with ABC during palliative CT can influence HRQoL, i.e., the greater the fatigue, the lower the OQoL of these patients; the opposite is also true, which can influence HRQoL positively or negatively.

In the evaluation of the Trait-Anxiety variable (STAI-T), we observed that women presented traits of anxiety already at baseline. The means of the instrument were close to 48.80 (CG: 48.31 ± 5.39, EG: 48.02 ± 7.62, SG: 48.38 ± 4.41) (*p* = 0.630). When the State Anxiety (STAI-S) was assessed, a baseline mean of 44.49/80 in the EG, 45.45/80 in the SG, and 44.79/80 in the CG were identified, with a slight decline in scores in the three groups, with no significant difference between the stages or between groups (T1, *p* = 0.810; T2, *p* = 0.840; T3, *p* = 0.640) (Table 7).

## 4. Discussion

Women with ABC are affected by the side effects of the disease itself, and its treatments, particularly palliative chemotherapy, which lead to changes in their daily lives and subsequently to cancer-related fatigue and anxiety throughout the therapeutic process, which directly interfere with HRQoL. The results of this research indicate poor quality of life in women with ABC starting chemotherapy. It is inferred that previous treatments and disease progression directly impact their overall quality of life. Thus, actions that can evaluate and use strategies to minimize these effects need to be part of the daily routine of health teams, aiming to improve survival and quality of life.

Of the adult women who started first- or second-line palliative chemotherapy for ABC during the study period, a low refusal to participate in the research may indicate that patients are interested in non-pharmacological therapies or interventions that can improve their HRQoL and/or place them as active participants in the treatment.

The profile of women with ABC in the present study is consistent between the groups and with that reported in other studies in Brazil and other countries, demonstrating that they are homogeneous groups. A Brazilian study [15] highlighted that the women had a mean age of 53.5 years, were married, and had children. Another study conducted in China [27] involved a population with a mean age of 49.7 years who were married, had studied up to elementary school/high school, were without comorbidities, and were inserted into the labor market. In all cases, cancer treatment negatively impacts women’s socioeconomic and marital spheres. It is perceived that the presence of a marital partner and children improves coping with cancer and its treatment, offering a stable support scenario for women [28]. It is noteworthy that the women in this study had a low income, ranging from one to three Brazilian minimum wages. This result is inferred to be related to the population and characteristics of the research setting, a hospital serving the Brazilian public healthcare system (SUS), which provides free care to the population.

Regarding clinical characteristics, the findings of the present study are similar to those performed with 176 Chinese women with ABC, most of whom (54%) had bone metastasis, and 51.1% were receiving first-line palliative chemotherapy instead of a combined regimen [5]. In short, even with a good prognosis and a high probability of survival in the first five years, this sample was composed of women with stage III and IV breast cancer. This leads to more aggressive treatments, with a worsening of the prognosis and an increase in symptoms during treatment, such as increased fatigue and anxiety and worsening HRQoL.

We compared the HRQoL of women with breast cancer undergoing palliative chemotherapy treated with or without non-pharmacological LA. The results showed that the overall scores of global quality of life, measured by the QLQ-C30, at the end of the 11 weeks, were significantly higher in the EG than in the SG and CG, demonstrating that HRQoL was influenced by the intervention. These results corroborate those of a study conducted in Taiwan [29], which included 450 breast cancer patients and compared patients who received and did not receive traditional Chinese medicine (TCM) practice. In that study, the quality of life in the TCM group was 81.60±11.67 points, which was significantly higher than the 78.80±13.10 points in the group without TCM (*p* < 0.05).

Beyond statistical significance, the magnitude of the effect observed in the EG represents a clinically relevant improvement, indicating a tangible impact on functional capacity, emotional well-being, and the reduction in adverse symptoms. The difference in scores demonstrates that laser auriculotherapy can provide noticeable and significant benefits to the quality of life of these women, highlighting its practical relevance in the context of palliative care. The use of LA reinforces the applicability of this NPT in clinical practice, particularly as a complementary support to conventional therapies.

Breast cancer patients commonly experience a variety of discomforts during treatment, such as physical changes, fatigue, pain, changes in eating, sleeping, and other problems, directly impacting their HRQoL. A study evaluating the effectiveness of auriculotherapy with needles (n = 143) or electroacupuncture (n = 145) in treating chronic musculoskeletal pain in cancer patients during ten sessions indicated significant results in reducing pain and the use of analgesics, and improved physical and mental quality of life compared to usual treatment (n = 72) [30]. Conversely, in the analysis of the subgroup (n = 165) of women with breast cancer from the same study, a significant improvement (*p* = 0.001) was observed in pain reduction through electroacupuncture. No statistically significant difference was observed between interventions for the mental health domain; however, both auriculoacupuncture with needles and electroacupuncture significantly improved HRQoL scores [31]. These data are in agreement with the results of the present study, in which the scores of physical functioning and personal performance increased in the EG and decreased in the other groups, showing a significant improvement in the OQoL scores of these women.

Importantly, the QoL score in the EG was higher than in the other groups after the interventions, suggesting a positive effect of NPT on HRQoL. A similar study conducted in Hong Kong [32] with women with breast cancer undergoing chemotherapy (n = 30), assessed the effects of electroacupuncture and auricular acupressure on QoL, evaluated by the FACT-B (specific questionnaire for QoL in breast cancer) over 14 weeks. Significant differences were observed, with an increase in HRQoL scores for patients who received NPT.

In another randomized study conducted on American women (n = 51) with breast cancer, the intervention was performed with systemic and auricular acupuncture and then compared with a sham technique during 12 sessions with two weekly applications. Among those who completed all stages, HRQoL measured with the FACT-G instrument showed a significant improvement in physical well-being (*p* = 0.03) in the intervention group, with no differences observed for the social/family, emotional, and functional well-being subscales [33].

Similarly, a clinical randomized trial was conducted with German female breast cancer survivors (n = 52) and compared the outcome of a 5-week auricular acupuncture intervention with needles. The outcomes were significantly better in those who underwent auricular acupuncture with needles (n = 26) than in those subjected to a single psychoeducational intervention (n = 26). Better sleep quality (*p* = 0.031), stress (*p* = 0.030), fatigue (*p* = 0.006), and anxiety (*p* = 0.001) were found in the group who underwent auricular acupuncture with needles. HRQoL scores were improved in the intervention group but were not statistically significant [34].

HRQoL is influenced by fatigue, which significantly affects the QoL of breast cancer patients during and after chemotherapy [35].

The results of this study showed that the CRF scores presented by women with ABC in the EG increased over the 11 weeks of evaluation and were significantly higher than the scores of women in the SG and CG, demonstrating a decrease in the reporting of this symptom and indicating the efficacy of LA. Our findings are in agreement with a study conducted in China, which evaluated the use of auriculotherapy in CRF in patients with lung cancer (n = 100) undergoing chemotherapy treatment. The study reported a significant improvement in participants who received the intervention with mustard seeds and magnetic beads, with no decrease in fatigue in the control group [36].

Similarly, an American pilot study analyzed the effect of auriculotherapy on different symptoms (pain, fatigue, and insomnia) in women (n = 31) with breast cancer undergoing chemotherapy treatment. Seeds were applied in the intervention group (n = 16) and a simulation of treatment in the same acupoints was performed in the control group (n = 15) for 4 weeks. The intervention group, experienced a reduction in fatigue (41%), pain (71%), insomnia (31%), and an improvement in the performance of daily activities (61%) [37].

The efficacy of auriculotherapy was also demonstrated in a study that evaluated the effects of chemotherapy in women with ovarian cancer (n = 65) [38]. In women in the control group, the mean scores measured by the symptom questionnaires increased, while the mean scores of the acupressure group decreased significantly in the three assessment periods (*p* = 0.03). After 4 weeks of auricular acupressure (time 2), fatigue scores decreased (*p* = 0.016). After 6 weeks of auricular acupressure (time 3), there was also a reduction in fatigue (*p* < 0.01).

When observing all the domains evaluated in this study, it becomes evident that LA demonstrated effects beyond statistical significance, with clinically relevant implications for patients’ quality of life. The improvement in physical, emotional, and functional well-being scores reflects a direct impact on the ability of women to face the daily challenges imposed by the disease and palliative treatment. These effects are particularly important for patients with advanced breast cancer, whose physical and emotional limitations often compromise their autonomy and overall well-being. The reduction in cancer-related fatigue observed in the experimental group is a clear example of a clinically significant benefit, as fatigue is a challenging symptom to manage and has a substantial negative impact on quality of life. The combination of efficacy, safety, and clinical impact highlights laser auriculotherapy as an essential tool for improving the quality of life in oncology patients.

Regarding the anxiety profile of the participants in the present study, the sample of women with breast cancer showed traces of anxiety (i.e., a stable variable that indicates a greater willingness to feel anxiety) prior to inclusion in the study. This condition is probably related to sociodemographic and clinical characteristics and the set of uncertainties generated by the diagnosis and treatment that breast cancer demands. A study performed in Taiwan [29] pointed out that patients with breast cancer present several discomfort symptoms during the treatment process, such as physical changes, fatigue and pain, eating, sleep, and other problems, including anxiety, depression, and anxiety associated with death, especially in women with advanced cancer.

Our data indicates that there was a slight, non-significant reduction in the state (particular moment or situation) of anxiety during the LA sessions, regardless of the group. This slight reduction may be linked to the conventional treatment (palliative chemotherapy) that the patient was receiving, as well as to an increased expectation of improvement in symptoms and/or quality of life, or even to the hope of having started a new therapy. A study conducted in South Korea to evaluate the effects of auricular acupressure on the anxiety of patients undergoing chemotherapy for breast cancer (experimental group, n = 30; control group, n = 30) showed a significant decrease in anxiety in the experimental group compared to the control group, confirming that auriculotherapy was an effective intervention to decrease the anxiety perceived by patients undergoing chemotherapy [39].

Another experimental study with auriculotherapy, carried out with Korean women (n = 60) undergoing chemotherapy for breast cancer, reported significant results in reducing anxiety (*p* ≤ 0.001) and increasing sleep (*p* ≤ 0.001) in the experimental group compared to the control group [39]. The reduction in anxiety was also significant (*p* = 0.020) in a feasibility study of the application of auriculotherapy during chemotherapy cycles for the treatment of breast cancer in North American women, in addition to significant reductions in the symptom burden and severity of nausea [40].

The absence of significant effects in reducing anxiety in the present study can be explained by several factors. First, the frequency of the sessions (once a week) and the timing of the questionnaire application (before the start of the session) may have been insufficient to produce a greater impact on anxiety levels or their measurement, as anxiety is a dynamic state often influenced by immediate situational and emotional factors. Additionally, anxiety in oncology patients can be influenced by multiple factors, such as diagnosis, prognosis, social support, and life events, which may overshadow the effects of specific interventions like auriculotherapy.

Another aspect to consider is the profile of the study participants, composed exclusively of women with advanced breast cancer undergoing palliative chemotherapy, a context that inherently carries a high emotional and psychological burden due to its reserved and uncertain prognosis. Previous studies that demonstrated a greater impact of auriculotherapy on anxiety often involved populations with moderate levels of anxiety or in less advanced stages of the disease, which could influence both the perception and response to the intervention [38,39,40].

Although the findings of this study did not show statistical significance for anxiety, they underscore the need to investigate complementary interventions, such as auriculotherapy, with greater intensity and frequency of application. Future studies could include more frequent measurements, such as assessing anxiety before and after each LA session, and evaluating the effects in specific subgroups, including patients with different baseline anxiety profiles or at varying stages of the disease. These efforts may place the findings in the broader context of existing literature and deepen the understanding of the therapeutic potential of auriculotherapy in managing anxiety in oncology patients.

Of the 123 women with breast cancer in the present study, none reported specific adverse effects and discomfort resulting from LA. The high rate of adherence to treatment with auriculotherapy indirectly indicated that blinding was successful due to the successful application of a placebo in the study. The findings of this study may provide valuable information and increase understanding of the therapeutic effect of LA in cancer patients and in chemotherapy treatment. The proposed treatment approach can be considered a non-invasive strategy for the management of cancer fatigue and anxiety.

The limitations of this study include a lack of follow-up evaluations beyond 11 weeks after the start of treatment; the application of the anxiety questionnaire in only three stages, since previous studies performed this assessment at each application or on a daily basis [39,40]; the participants are from a single center (thus limiting generalization of the results); and the inability to adopt a double-blind approach because the researcher could not be blinded considering the application of the intervention.

For future studies, it is recommended to investigate the efficacy of laser auriculotherapy over longer periods, including medium- and long-term follow-ups to evaluate the maintenance of the benefits observed in HRQoL. Longitudinal studies could also explore the applicability of this technique to other common cancer symptoms, such as pain, nausea, and sleep disorders. Investigations with diverse population subgroups, including different age ranges, comorbidities, and cancer stages, as well as men with various types of cancer, are recommended. Multicenter trials with representative samples could strengthen the findings and facilitate their implementation in public health policies. Additionally, it is crucial to explore the technique’s impact on reducing hospitalizations and emergency visits, reinforcing its role as an integrative practice in palliative care.

## 5. Conclusions

This study indicates that an intervention of LA for 10 weeks improved HRQoL and cancer-related fatigue in women with ABC undergoing palliative chemotherapy from the 5th week of the application of the therapy. However, LA did not interfere with anxiety levels during treatment. A longer duration of auriculotherapy and more frequent measurements may be followed in future trials to determine improvements in anxiety in these patients.

The intervention of LA is an economical and feasible practice that demonstrates the potential to reduce cancer-related fatigue, a symptom that is difficult to control, in a complementary way. The low-level laser used to perform auriculotherapy was effective in the complementary treatment of women with cancer, causing no discomfort or pain once a week, and it was not necessary to stimulate the points throughout the day or during treatment. Thus, the therapy did not present adverse events and was safe, low-cost, and easily applicable.

## Figures and Tables

**Figure 1 healthcare-13-00166-f001:**
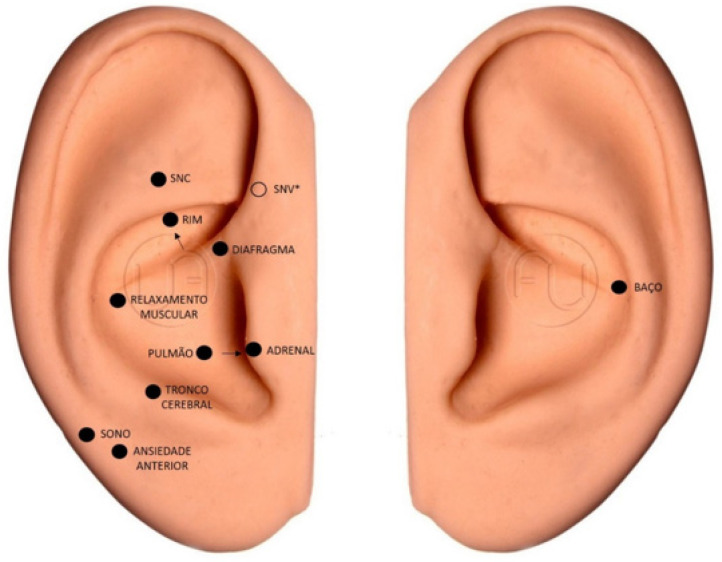
Location of acupoints in the dominant ear and left ear. Prepared by the authors (2023). * Point located on the inside of the ear. Legend: SNC = CNS—Central nervous system; SNV = NNS—Neurovegetative Nervous System; Rim = Kidney; Diafragma = Diaphragm; Relaxamento Muscular = Muscle Relaxation; Pulmão = Lung; Adrenal = Adrenals; Tronco Cerebral = Brainstem; Sono = Sleep; Ansiedade Anterior = Anxiety; Baço = Spleen.

**Figure 2 healthcare-13-00166-f002:**
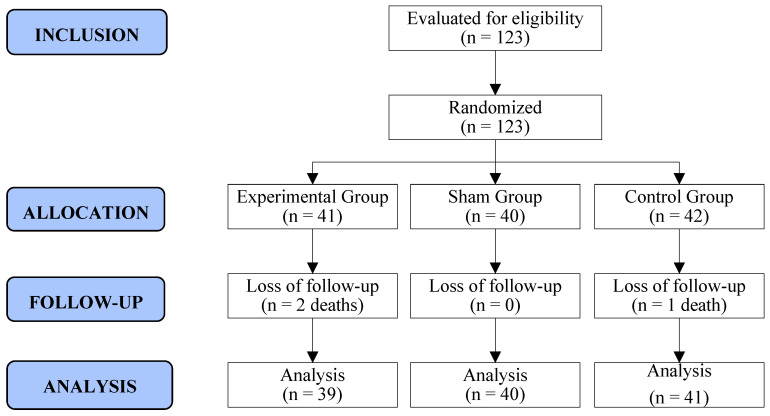
Flowchart of recruitment and allocation of study participants. Curitiba, Paraná, Brazil. Prepared by the authors (2024).

**Figure 3 healthcare-13-00166-f003:**
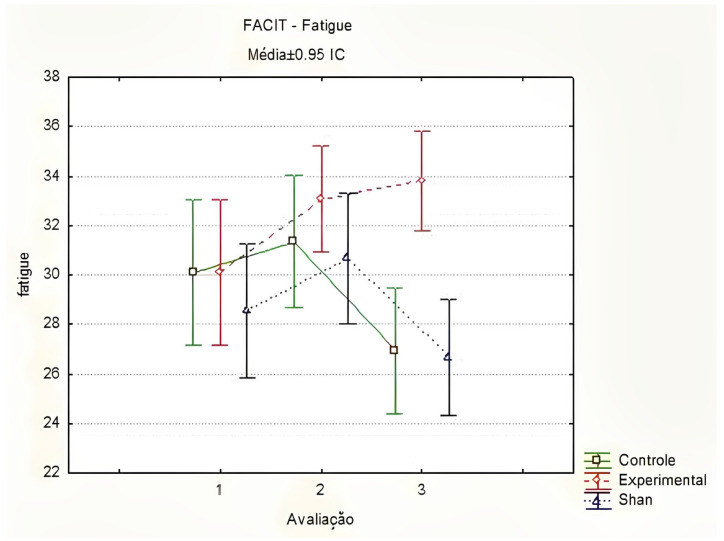
Comparison of fatigue by FACIT-Fatigue between groups and stages, using the Friedman method. Curitiba, Paraná, Brazil. Legend: Médias = Médias ± 0.95 IC = Means ± 95%CI; Controle = Control; Experimental = Experimental; Shan = Sham; Avaliação = Evaluation. Prepared by the authors (2024).

**Table 1 healthcare-13-00166-t001:** Randomization by block and corresponding group.

Block	Card Numbering	Group
1, 2, 3 and 4	1, 2, 3, 4, 5, 6, 7, 8, 9, 10	EG
	11, 12, 13, 14, 15, 16, 17, 18, 19, 20	SG
	21, 22, 23, 24, 25, 26, 27, 28, 29, 30	CG

Prepared by the authors (2024). Legend: EG—experimental group; SG—sham Group; CG—control group.

**Table 2 healthcare-13-00166-t002:** Sociodemographic and clinical characteristics of patients undergoing palliative chemotherapy for advanced breast cancer randomized into experimental, sham, and control groups. Curitiba, Paraná, Brazil.

Variable	Experimental Group (n = 41)		Group Sham (n = 40)		Control Group (n = 42)		*p*
	n	%	n	%	n	%	
**Age in years (mean ± SD)**	55.17 ± 11.06		50.88 ± 10.50		53.64 ± 12.01		0.260
**Karnofsky (mean)**	69		69		70		0.82
**Marital status**							
Single	14	34.15	13	32.50	10	23.81	0.130
Married/stable relation	17	41.46	22	55	28	66.67	
Separated/divorced	8	19.51	4	10	1	2.38	
Widow	2	4.88	1	2.50	3	7.14	
**Schooling**							
Functionally illiterate	3	7.32	1	2.50	0	0	
4 to 7 years schooling	13	31.71	26	65	20	47.62	** *0.032 ** **
8 to 10 years schooling	19	46.34	11	27.50	15	35.71	
11 or more years schooling	6	14.63	2	5	7	16.67	
**Employment**							
Active	22	53.66	28	70	16	38.10	** *0.015 ** **
Retired	10	24.39	5	12.50	5	11.90	
Domestic	5	12.20	5	12.50	6	14.29	
Unemployed	4	9.76	2	5	15	35.71	
**Income ****							
No income	0	0	3	7.50	0	0	0.920
Up to USD 220.00	0	0	6	15	1	2.38	
USD 220.00 to 660.00	36	87.80	27	67.50	37	88.10	
USD 880.00 to 2200.00	4	9.76	3	7.50	4	9.52	
USD 2420.00 to 4400.00	1	2.44	1	2.50	0	0	
**No. of children**							
None	7	17.07	6	15	5	11.9	
Only 1 child	22	53.66	17	42.50	18	42.86	0.520
2 to 3 children	10	24.39	15	37.50	13	30.95	
More than 3 children	2	4.88	2	5	6	14.29	
**Religious belief**							
Present	40	97.56	37	92.50	41	97.62	0.410
**Comorbidities**							
Present	20	48.78	21	52.50	10	23.81	** *0.016 ** **
**Number of metastases**							
1	14	34.15	14	35	15	35.71	0.910
2	17	41.46	18	45	17	40.48	
3	9	21.95	8	20	8	19.05	
4	1	2.44	0	0	2	4.76	

Prepared by the authors (2024). Legend: n: number of participants. Note: *** *p: <0.05 statistical significance***; ** USD 220.00 is the approximate value of the Brazilian minimum wage during the survey period.

**Table 3 healthcare-13-00166-t003:** Comparison between stages of the health-related quality of life scores of the Quality of Life Questionnaire—Core 30, functional scale of women with advanced breast cancer undergoing palliative chemotherapy in the three stages of the research. Curitiba, Paraná, Brazil.

Global Quality of Life	Baseline			Assessment 2			Assessment 3			Comparison Between Stages **
Group	n	Mean	SD	n	Mean	SD	n	Mean	SD	** *p* **
Experimental	41	31.71	12.53	41	32.11	10.47	39	47.44	12.56	** *<0.001 ** **
Sham	39	28.85	15.51	40	20.63	11.48	40	15.63	6.85	** *<0.001 ** **
Control	42	25.79	12.73	42	20.63	10.12	41	12.40	10.72	** *<0.001 ** **
**Physical Functioning**	**Baseline**			**Assessment 2**			**Assessment 3**			
Group	n	Mean	SD	n	Mean	SD	n	Mean	SD	** *p* **
Experimental	41	47.37	21.87	41	50.61	17.17	39	51.45	17.83	0.580
Sham	39	46.15	21.38	40	50.67	21.07	39	35.04	17.35	** *0.005 ** **
Control	42	46.03	21.33	42	51.43	21.35	41	40.65	19.25	0.190
**Personal performance**	**Baseline**			**Assessment 2**			**Assessment 3**			
Group	n	Mean	SD	n	Mean	SD	n	Mean	SD	** *p* **
Experimental	41	46.75	26.67	41	49.59	25.95	39	44.87	20.29	0.650
Sham	39	38.46	16.29	40	42.08	16.45	39	41.03	17.46	0.390
Control	42	42.46	18.48	42	42.86	18.09	41	40.65	18.28	0.900
**Emotional functioning**	**Baseline**			**Assessment 2**			**Assessment 3**			
Group	n	Mean	SD	n	Mean	SD	n	Mean	SD	** *p* **
Experimental	41	45.73	19.01	41	47.97	14.70	39	55.13	15.95	** *0.007 ** **
Sham	39	51.71	17.44	40	52.29	16.12	39	43.59	17.57	** *0.053 ** **
Control	42	53.17	17.36	42	52.38	13.56	41	45.33	22.36	0.530
**Cognitive functioning**	**Baseline**			**Assessment 2**			**Assessment 3**			
Group	n	Mean	SD	n	Mean	SD	n	Mean	SD	** *p* **
Experimental	41	39.02	20.62	41	43.09	21.07	39	39.74	13.03	0.850
Sham	39	48.29	20.16	40	48.75	14.81	39	22.22	22.40	** *<0* ** **.** ** *001 ** **
Control	42	48.81	23.10	42	45.24	14.39	41	31.71	27.59	** *0.008 ** **
**Social Functioning**	**Baseline**			**Assessment 2**			**Assessment 3**			
Group	n	Mean	SD	n	Mean	SD	n	Mean	SD	** *p* **
Experimental	41	41.87	15.86	41	38.62	15.11	39	29.91	16.75	** *0.006 ** **
Sham	39	44.44	13.96	40	44.58	12.74	39	25.21	17.47	** *<0.001* **
Control	42	42.46	12.86	42	43.65	13.25	41	32.52	21.39	** *0.003 ** **

Prepared by the authors (2024). Legend: n: number of participants; SD—standard deviation. Note: **** p: <0.05 statistical significance***; ** Friedman test.

**Table 4 healthcare-13-00166-t004:** Comparison between stages of the health-related quality of life scores of the Quality of Life Questionnaire—Core 30, functional scale of women with advanced breast cancer undergoing palliative chemotherapy in the three stages of the research. Curitiba, Paraná, Brazil.

Fatigue	Baseline			Assessment 2			Assessment 3			Comparison Between Stages **
Group	n	Mean	SD	n	Mean	SD	n	Mean	SD	** *p* **
Experimental	41	56.91	16.70	41	50.41	13.85	39	48.72	15.21	** *0.020 ** **
Sham	39	58.40	17.24	40	55.28	17.16	39	68.38	16.82	** *0.020 ** **
Control	42	55.03	20.23	42	55.03	18.03	41	69.11	17.66	** *0.001 ** **
**Nausea and Vomiting**	**Baseline**			**Assessment 2**			**Assessment 3**			
Group	n	Mean	SD	n	Mean	SD	n	Mean	SD	** *p* **
Experimental	39	61.54	23.62	41	62.60	18.55	39	79.49	24.92	** *<0.001 ** **
Sham	39	43.59	21.84	40	45.83	22.25	39	70.09	34.87	** *<0.001 ** **
Control	42	45.24	24.22	42	46.83	20.90	41	61.79	32.97	** *0.002 ** **
**Pain**	**Baseline**			**Assessment 2**			**Assessment 3**			
Group	n	Mean	SD	n	Mean	SD	n	Mean	SD	** *p* **
Experimental	41	55.28	28.49	40	59.17	24.45	37	61.26	24.23	0.610
Sham	39	56.41	26.66	39	52.99	19.82	39	81.20	23.93	** *<0.001 ** **
Control	42	58.73	29.27	40	56.67	20.25	41	68.29	26.82	** *0.012 ** **
**Dyspnea**	**Baseline**			**Assessment 2**			**Assessment 3**			
Group	n	Mean	SD	n	Mean	SD	n	Mean	SD	** *p* **
Experimental	41	53.66	26.75	41	55.28	17.65	39	57.26	25.30	0.700
Sham	39	55.56	23.36	40	50.00	22.65	39	75.21	23.84	** *<0.001 ** **
Control	42	56.35	22.68	42	50.79	23.56	41	66.67	28.87	0.058
**Insomnia**	**Baseline**			**Assessment 2**			**Assessment 3**			
Group	n	Mean	SD	n	Mean	SD	n	Mean	SD	** *p* **
Experimental	39	63.25	29.41	41	51.22	25.92	39	53.85	23.71	0.090
Sham	39	82.05	26.32	40	73.33	28.44	35	75.24	23.35	** *0.037 ** **
Control	42	71.43	32.57	42	68.25	29.40	39	73.50	29.79	0.130
**Loss of Appetite**	**Baseline**			**Assessment 2**			**Assessment 3**			
Group	n	Mean	SD	n	Mean	SD	n	Mean	SD	** *p* **
Experimental	39	61.54	23.62	41	62.60	18.55	39	79.49	24.92	** *<0.001 ** **
Sham	39	43.59	21.84	40	45.83	22.25	39	70.09	34.87	** *<0.001 ** **
Control	42	45.24	24.22	42	46.83	20.90	41	61.79	32.97	** *<0.001 ** **
**Constipation**	**Baseline**			**Assessment 2**			**Assessment 3**			
Group	n	Mean	SD	n	Mean	SD	n	Mean	SD	** *p* **
Experimental	40	38.33	31.62	38	44.74	26.02	38	64.04	29.39	** *0.001 ** **
Sham	39	56.41	24.37	40	54.17	24.68	39	47.01	28.32	0.270
Control	42	61.11	24.32	42	57.94	25.57	41	50.41	28.01	0.210
**Diarrhea**	**Baseline**			**Assessment 2**			**Assessment 3**			
Group	n	Mean	SD	n	Mean	SD	n	Mean	SD	** *p* **
Experimental	40	61.67	31.62	38	55.26	26.02	38	35.96	29.39	** *0.001 ** **
Sham	39	43.59	24.37	40	45.83	24.68	39	52.99	28.32	0.270
Control	42	38.89	24.32	42	42.06	25.57	41	49.59	28.01	0.210
**Financial Difficulties**	**Baseline**			**Assessment 2**			**Assessment 3**			
Group	n	Mean	SD	n	Mean	SD	n	Mean	SD	** *p* **
Experimental	41	64.23	36.05	40	60.00	32.20	39	68.38	35.83	0.400
Sham	38	61.40	32.44	39	67.52	30.09	39	69.23	29.99	0.250
Control	40	59.17	35.80	39	66.67	31.53	40	75.83	25.02	0.057

Prepared by the authors (2024). Legend: n: number of participants; SD—standard deviation. Note: **** p: <0.05 statistical significance***; ** Friedman test.

**Table 5 healthcare-13-00166-t005:** Comparison between stages of the health-related quality of life scores of the Functional Assessment of Cancer Therapy-General, Physical, Social, Emotional, Functional and Fatigue Scale of women with advanced breast cancer undergoing palliative chemotherapy in the three stages of the research. Curitiba, Paraná, Brazil.

Physical Well-Being	Baseline			Assessment 2			Assessment 3			Comparison Between Stages **
Group	n	Mean	SD	n	Mean	SD	n	Mean	SD	** *p* **
Experimental	41	14.87	6.87	41	16.04	5.48	39	16.60	3.24	0.420
Sham	39	15.79	5.40	40	16.56	5.11	39	12.51	4.30	** *<0.001 ** **
Control	42	15.48	5.56	42	16.29	4.97	40	14.30	4.8	0.160
**Social/Family Welfare**	**Baseline**			**Assessment 2**			**Assessment 3**			
Group	n	Mean	SD	n	Mean	SD	n	Mean	SD	** *p* **
Experimental	41	14.68	4.09	41	14.88	3.81	39	14.49	3.23	0.780
Sham	39	11.90	4.85	40	12.81	4.46	39	12.73	3.64	0.070
Control	42	12.64	4.80	42	12.93	4.43	40	12.19	5.2	0.090
**Emotional Well-Being**	**Baseline**			**Assessment 2**			**Assessment 3**			
Group	n	Mean	SD	n	Mean	SD	n	Mean	SD	** *p* **
Experimental	41	13.65	5.17	41	14.88	4.25	39	15.70	2.34	0.200
Sham	39	14.63	3.40	40	14.53	2.68	39	13.13	3.38	0.060
Control	42	15.83	4.16	42	14.40	3.04	40	13.41	3.7	0.060
**Functional Well-Being**	**Baseline**			**Assessment 2**			**Assessment 3**			
Group	n	Mean	SD	n	Mean	SD	n	Mean	SD	** *p* **
Experimental	41	12.31	4.31	41	13.41	4.35	39	12.87	3.16	0.110
Sham	39	8.95	4.03	40	10.08	4.08	39	8.87	2.54	0.270
Control	42	9.60	4.82	42	10.26	4.32	40	9.53	4.0	0.580
**TOI**	**Baseline**			**Assessment 2**			**Assessment 3**			
Group	n	Mean	SD	n	Mean	SD	n	Mean	SD	** *p* **
Experimental	41	55.99	19.01	41	61.83	16.29	39	63.32	10.40	0.100
Sham	39	54.36	15.51	40	58.08	14.70	39	47.69	11.71	0.025
Control	42	55.88	16.60	42	57.91	15.80	40	50.78	13.30	0.340
**FACT-G**	**Baseline**			**Assessment 2**			**Assessment 3**			
Group	n	Mean	SD	n	Mean	SD	n	Mean	SD	** *p* **
Experimental	41	55.51	14.93	41	59.22	13.36	39	59.66	8.36	** *0.002 ** **
Sham	39	51.27	11.05	40	53.96	10.28	39	47.25	7.35	** *<0.001 ** **
Control	42	53.54	14.37	42	53.89	11.94	40	49.43	11.10	0.030
**FACIT-F**	**Baseline**			**Assessment 2**			**Assessment 3**			
Group	n	Mean	SD	n	Mean	SD	n	Mean	SD	** *p* **
Experimental	41	83.32	23.50	41	91.58	20.29	39	93.51	13.07	** *0.011 ** **
Sham	39	80.89	17.41	40	85.41	16.66	39	73.55	13.14	** *0.003 ** **
Control	42	84.35	21.21	42	85.25	19.17	40	76.38	16.86	0.027

Prepared by the authors (2024). Legend: n: number of participants; SD—standard deviation; FACIT: Fatigue-Functional Assessment of Chronic Illness Therapy-Fatigue. *p:* significance; FACT-G-Functional Assessment of Cancer Therapy-General; TOI = Trial Outcome Index (composed of the sum of the scores of physical well-being, functional well-being, and fatigue subscale). Note: **** p: <0.05 statistically significant***; ** Friedman test.

**Table 6 healthcare-13-00166-t006:** Correlation between the fatigue subscale score (FACIT-FS) and FACT-G and FACIT-F General Quality of Life scores in the experimental, sham, and control groups. Curitiba, Paraná, Brazil.

Control Group	Baseline			T2			T3		
	n	Spearman	*p*	n	Spearman	*p*	n	Spearman	*p*
FACIT-FS vs. FACT-G	42	0.63	0.000009	42	0.70	0.000000	40	0.67	0.000002
FACIT-FS vs. FACIT-F	42	0.89	0.000001	42	0.91	0.000000	40	0.85	0.0000001
**Sham Group**	**n**	**Spearman**	** *p* **	**n**	**Spearman**	** *p* **	**n**	**Spearman**	** *p* **
FACIT-FS vs. FACT-G	39	0.52	0.0005	40	0.58	0.00007	39	0.56	0.000219
FACIT-FS vs. FACIT-F	39	0.87	0.000001	40	0.85	0.000001	39	0.89	0.000001
**Experiment Group**	**n**	**Spearman**	** *p* **	**n**	**Spearman**	** *p* **	**n**	**Spearman**	** *p* **
FACIT-FS vs. FACT-G	41	0.78	0.000000	41	0.69	0.000001	39	0.62	0.000002
FACIT-FS vs. FACIT-F	41	0.89	0.000000	41	0.88	0.000000	39	0.86	0.0000001

Prepared by the authors (2024). Legend: n: number of participants; *p*: significance; FACT-G: Functional Assessment of Cancer Therapy-General; FACIT-F: Functional Assessment of Chronic Illness Therapy-Fatigue; FACIT-FS: Subscale Fatigue.

**Table 7 healthcare-13-00166-t007:** Comparison of state anxiety (STAI-S) between stages and between groups. Curitiba, Paraná, Brazil.

Group	Baseline				2nd Assessment				3rd Assessment				*p **
	n	Mean ± SD	Min.	Max.	n	Mean ± SD	Min.	Max.	n	Mean ± SD	Min.	Max.	
Control Group	42	44.79 ± 5.63	32	60	41	43.81 ± 5.82	31	57	40	42.66 ± 6.13	30	55	0.300
Experimental Group	41	44.49 ± 5.67	30	56	41	43.24 ± 4.22	36	50	39	43.31 ± 4.65	35	53	0.990
Sham Group	40	45.45 ± 6.47	32	60	41	43.03 ± 5.91	31	57	41	41.80 ± 6.33	30	55	0.240
Total	123	44.9 ± 5.89	30	60	123	43.37 ± 5.34	31	57	120	42.58 ± 5.75	30	55	

Prepared by the authors (2024). Legend: n: number of participants; SD: standard deviation; Min.: minimum; Max.: maximum; p: significance. Note: * *p*: <0.05 statistically significant.

## Data Availability

Raw data supporting the conclusions of this article will be made available by the authors on request.

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
