# Peer review of "Effect of Laser Auriculotherapy on Quality of Life, Fatigue, and Anxiety in Women with Advanced Breast Cancer: Randomized Clinical Trial"

_healthcare, 2025, doi:10.3390/healthcare13020166_

Round 1

Reviewer 1 Report

Comments and Suggestions for Authors

Thanks to the authors for such interesting research on improving the health-related quality of life of woman with advanced breast cancer, via assessing the effect of laser auriculotherapy.

After a careful appraisal of the paper, I acknowledge the effort put by the authors in properly conduct the research, including the literature review, methodology, and data collection by using proven data-gathering tools. I am having, however, some concerns regarding the statistical tools and further analyses conducted by the authors in Section 3, that led for the reconsideration of the paper, hoping authors will be able to address them thoughtfully.

1.      [Paragraph starting in line 191] Authors introduce the instruments implemented for undertaking patients’ questionnaires. Among them, QLQ-C30, which is divided into four domains. Adding them all up, a general score ranging between 0 to 100 can be generated. I am puzzled here as no data about this composite score is presented in the paper whatsoever. If authors refer to Global Quality of Life item shown in table 2, why are the means so low? (out of 100?)

2.      [Paragraph starting in line 230] Authors explain the statistical tests used to assess the scores obtained from HRQoL. Within the tests listed, authors refer to the non-parametric versions, namely Kruskal-Wallis (for comparing variables with more than two categories), Mann-Whitney test (when dealing with variables with two categories only), and Wilcoxon (rank) test (when comparing between evaluations.) Also, authors explicitly mention a level of significance alpha = 0.05. Assuming the normality in the data, the correct statistical test for the experimental design proposed and implemented by the authors in this research, would be some adaptation of the One-Way / Two-Way ANOVA within/between subjects. Although there is nothing incorrect in the use of non-parametric tests in this research, it seems that authors forgot the problem of pairwise comparisons that inflate the probability of error type I (which is controlled in the parametric version of the test mentioned previously). Thus, the need to correct the p-value using well-known technics such as Bonferroni Correction.

3.      Results section is sometimes hard to follow. Please, see my comments in the next points.

4.      Table 1(demographics): It seems not too relevant for the core of the study presented and could be summarized in a different way.

5.      Table 2: following point 2 above, this table reports p-values based on non-parametric techniques. However, authors draw conclusions from it by referencing to the values of mean and standard deviations, and not based on medians and quartiles.

6.      Table 2: “Personal Performance” item is not discussed by the authors.

7.      Tabe 2 (reference line 284): authors argue “…significant increase (p<0.001) between T2 and T3.” As mentioned earlier, it is not clear how the authors are performing pairwise comparisons (where did this p-value come from?).

8.      Table 2: Overall results appear to be unable to uniquely explain a better performance of the experimental group (please, see my general note at the end.)

9.      Table 3: Similar comment as point 8 above. Besides Financial Difficulties, the significance of the variables appears to alternate among the groups. As comment aside, should this variable (Financial Difficulties) even need to be considered here?

10. Table 3: there is something wrong with the numbers presented, as some of the standard deviations shown are repeated among groups and times.

11. Table 4: please check the conclusions drawn from this table. As an example, in the paragraph starting in line 334, a comparison within groups is made, missing the important comparison among groups.

12. Graph in Figure 3. This graph is self-explanatory, as it clearly shows the difference between the experimental group against CG and SH. However, it appears that there is an inconsistency between this graph showing medians and all the tables showing means and standards deviations.

General note: authors should report confidence intervals, alongside appropriate statistics and effect sizes according to the tests performed. I am especially worried about the overall purpose of this research as, according to the results presented, the experimental group appears to perform better than the control and shame groups. However, I couldn’t see any argument from the authors if this better performance is clinically significant.

Author Response

321 / 5,000

Dear reviewer,   Below are the adjustments made based on your comments. We appreciate your feedback to improve the quality of the article. All changes to the text have been highlighted in yellow to make them easier to identify. We remain at your disposal for any further clarification.   The comments are attached.

Reviewer 2 Report

Comments and Suggestions for Authors

Thank you for conducting this well-designed study exploring a non-pharmaceutical intervention for common problems in advanced breast cancer. We know that this is something which is attractive to patients and we also know that the evidence base for CAM in cancer is small and of variable quality.

I would suggest that the discussion looks in a little more detail at the differences in results between the experimental group and the sham condition in addition to the differences between experimental and control group.

It would be useful to know whether there were any constraints on the control group eg could they have been having acupuncture outside the study.

Comments on the Quality of English Language

There are some instances of wording which are less clear than is optimal eg line 93 refers to 'discontinuity criteria' - is this the same as exclusion criteria?

Author Response

Dear Reviewer,

Below are the adjustments made according to your comments. We appreciate your feedback to enhance the quality of the article. All changes in the text have been highlighted in yellow to facilitate their identification. We remain available for further clarifications.

Reviewer 3 Report

Comments and Suggestions for Authors

Your manuscript describes interesting results of using LA as an alternative therapy for unmet medical needs, specifically fatigue and anxiety, that exist in female breast cancer patients who are undergoing treatment, and this is of high clinical significance.

I point out the following points for revision of this manuscript.

Major Comments

1. Patient Background Heterogeneity: The study identifies considerable differences in patient baseline characteristics across groups, such as variations in KPS, educational level, employment status, and comorbidities. The authors should consider either implementing statistical adjustments to mitigate these baseline discrepancies or provide a detailed discussion on how these potential confounders may influence the study's outcomes.

2. Baseline Value Variations: Significant differences in baseline values across groups may substantially impact the interpretation of subsequent numerical changes. It is recommended that the authors perform a comprehensive analysis of changes relative to baseline and adopt statistical methods capable of addressing these initial imbalances.

3. Clinical Significance: The manuscript needs to move beyond statistical significance to a more nuanced discussion of clinical effect sizes. This involves clearly articulating the magnitude of intervention effects and elaborating on the clinical relevance of observed changes.

4. Mechanistic Discussion: The authors are encouraged to expand on the mechanisms underlying the intervention's impact on fatigue and explore potential explanations for the lack of observed effects on anxiety. Additionally, situating these findings within the context of existing literature would enhance the manuscript's depth.

5. Future Research Directions: The conclusion or discussion section should outline future research pathways, including studies on the long-term efficacy of the intervention and its applicability across diverse population subgroups.

Minor Comments:

1. Data Reporting Inconsistency: Table 1 contains an income category gap that requires addressing to ensure complete data representation.

2. Figure 3 Visualization Improvements: The visualization in Figure 3 requires refinement. To improve clarity, consider adopting a more distinguishable color scheme (e.g., red for the experimental group, blue for the sham group, and green for the control group). Additionally, optimizing axis label positions would enhance legibility.

Author Response

(The authors gave the same response as above.)

Round 2

Reviewer 3 Report

Comments and Suggestions for Authors

Thank you for your response to my comments. After reviewing your revisions, I am pleased to accept your manuscript.